# Factors Associated with Disagreement of Fibrosis Stages between 2D-Shear Wave Elastography and Transient Elastography in Chronic Hepatitis B

**DOI:** 10.3390/v15040846

**Published:** 2023-03-26

**Authors:** Fernanda Gdalevici Miodownik, Ana Carolina Cardoso, Leticia Cancella Nabuco, Cibele Franz, Renata Perez, Cristiane Alves Villela-Nogueira

**Affiliations:** 1Hepatology Division, School of Medicine, Federal University of Rio de Janeiro, Rio de Janeiro 21941-617, Brazil; 2Gastroenterology Department, Federal University of the State of Rio de Janeiro, Rio de Janeiro 20270-004, Brazil; 3Gastroenterology Department, University of the State of Rio de Janeiro, Rio de Janeiro 20950-003, Brazil; 4D’Or Institute for Research and Education (IDOR), Rio de Janeiro 22281-100, Brazil

**Keywords:** diabetes mellitus, liver elastography, fatty liver, chronic hepatitis B, cirrhosis

## Abstract

Introduction and objectives: The agreement of elastography techniques in chronic Hepatitis B (CHB) needs evaluation. We aimed to evaluate, in CHB, the agreement between transient elastography (TE) and two-dimensional shear wave elastography (2D-SWE), analyzing the factors related to the disagreement of measures. Materials and methods: CHB patients underwent liver stiffness measures with both TE and 2D-SWE on the same day. For concordance analysis, we defined liver fibrosis as F0/1 vs. F ≥ 2, F0/1-F2 vs. F ≥ 3 and F0/1-F2-F3 vs. F4 for both methods. Logistic regression analysis was used to identify the variables independently associated with the disagreement between methods. RESULTS: A total of 150 patients were enrolled. Liver fibrosis categorization according to TE was: F0-F1 = 73 (50.4%), F ≥ 2 = 40 (27.6%), F ≥ 3 = 21 (14.5%) and F4 = 11 (7.6%), and according to 2D-SWE was: F0/F1 = 113 (77.9%), F ≥ 2 = 32 (22.1%), F≥ 3 = 25 (17.2%) and F4 = 11 (7.6%). It was observed that 20.0% of the sample had steatosis (CAP≥ 275 dB/m). TE and SD-SWE estimated equal fibrosis stages in 79.3% of cases. Spearman's correlation coefficient was 0.71 (*p* < 0.01). Kappa values for F ≥ 2, F ≥ 3 and F = 4 were: 0.78, *p* < 0.001; 0.73, *p* < 0.001; and 0.64, *p* < 0.001, respectively. Diabetes mellitus (DM) (OR 5.04; 95%CI: 1.89–13.3; *p* < 0.001) and antiviral treatment (OR 6.79; 95%CI: 2.33–19.83; *p* < 0.001) were independently associated with discordance between both methods. Conclusions: In CHB, there is strong correlation and good agreement between TE and 2D-SWE in identifying fibrosis stages. Diabetes mellitus and antiviral therapy may impact the agreement of stiffness measures obtained with these elastographic methods.

## 1. Introduction

Hepatitis B infection is a world health problem. In 2015, the World Health Organization estimated that 257 million people live with chronic hepatitis B infection [1]. The health impact of HBV infection has increased over time as individuals with chronic infection have aged and developed HBV-related complications. Chronic HBV infection is a dynamic process reflecting the interaction between HBV replication and the host immune response, and not all patients with chronic HBV infection have chronic hepatitis (CHB). Almost 40% of individuals with CHB progress to cirrhosis, liver failure and hepatocellular carcinoma (HCC) [2]. 

Estimating liver fibrosis stage is essential for managing CHB and advanced fibrosis complications [3,4]. Liver biopsy is the gold standard for the assessment of liver fibrosis. However, it is an invasive procedure associated with severe complications—especially local pain and bleeding—in 1–3% of cases and a mortality rate of approximately 1 in 10,000 [5,6]. Furthermore, the accuracy of liver biopsy is influenced by sample size and length, and intra- and interobserver variation are presented in the estimation of liver fibrosis stage [7].

Assessing the severity of liver disease is essential to identify patients for treatment and HCC surveillance. Early diagnosis of significant fibrosis and treatment of CHB could delay or even prevent the development of cirrhosis and hepatocellular carcinoma in CHB patients. In clinical practice, using a noninvasive method for accurately detecting liver fibrosis is very important, as such tools can be used to monitor the progression of liver fibrosis over time. In patients with hepatitis B, unlike hepatitis C, fibrosis degree is relevant to treatment decisions [3,4].

Noninvasive methods for liver fibrosis evaluation were developed to reduce the need for liver biopsy. They have applicabilities of over 95% (in patients who are not morbidly obese), provide real-time results and are fast procedures with short learning curves for operators [8]. However, liver stiffness is a physical property of the tissue, which depends not only on the amount of liver fibrosis but can be influenced by several other factors. Therefore, liver stiffness measurement (LSM) results can overestimate fibrosis in cases of inflammation, obstructive cholestasis, food ingestion, exercise, or venous congestion. These should be carefully excluded to avoid misdiagnosis [9].

Transient elastography using Fibroscan^®^ (Echosens, Paris, France) was the first elastography method developed. Owing to its applicability in clinical practice and validation, TE is recommended for noninvasively staging hepatic fibrosis in different guidelines [3,4]. The ability to quantify steatosis by measuring ultrasonic attenuation of the echo wave has been implemented in the FibroScan^®^ device; this has been named the controlled attenuation parameter (CAP). In the past, transient elastography (TE) was best validated for hepatitis C chronic infection (HCV); however, studies have shown similar results in HBV patients [10]. 

Two-dimensional shear wave elastography (2D-SWE) is a novel noninvasive method for staging liver fibrosis that combines shear waves bound on a conventional ultrasound (US) probe, providing a real-time quantitative map of liver tissue stiffness. The B-mode image also enables the evaluation of liver morphology and hepatic vessels. In addition, it can be used in patients with ascites [11,12,13]. 

Studies focusing on hepatitis B and 2D-SWE are rare, and most of them are performed with Asian patients who usually have a lower BMI [14,15,16,17] than the Caucasian population. Some of these previous studies compared the accuracy of TE and 2D-SWE. However, the factors influencing the discordance of liver stiffness measurements between these two methods in CHB individuals have not been evaluated. 

Thus, this study aimed to evaluate the agreement between TE and 2D-SWE in patients with chronic hepatitis B and analyze the factors related to the disagreement of measures between these two elastography methods. 

## 2. Materials and Methods

### 2.1. Patients and Study Design

This was a single-center, cross-sectional study with prospective inclusion. Between November 2016 and February 2019, consecutive patients with CHB were recruited. The inclusion criteria were a positive hepatitis B surface antigen test for more than six months and an age equal to or above 18 years. The exclusion criteria were hepatitis C or HIV coinfection, hepatocellular carcinoma (HCC), autoimmune and cholestatic diseases, hemochromatosis, alcohol consumption of over 20 g/day for females and 30 g/day for males, alanine aminotransferase over five times the upper limit of normal or biochemical cholestasis, pregnancy, and impossibility of obtaining elastography measures or unreliable elastography results. 

All included patients underwent liver stiffness measures with TE and 2D-SWE on the same day, with at least 3 h of fasting. Two experienced operators performed all the exams, blinded to clinical data and their counterpart's elastography results. In the final analysis, the patients included had valid liver stiffness measures obtained via elastographic methods. Anthropometric measures (body weight and height, body mass index/BMI and waist circumference), history of systemic arterial hypertension, diabetes mellitus and dyslipidemia were registered, as were those under antiviral therapy for chronic HBV infection. The maximum interval between elastography and laboratory tests was 60 days. The following laboratory tests were performed and included in the analysis for all patients: alanine aminotransferase (ALT), aspartate aminotransferase (AST), γ-glutamyltransferase (γGT), alkaline phosphatase, platelet count, glucose, HbA1c, albumin, prothrombin activity levels, total bilirubin, total cholesterol and triglycerides. Written informed consent was obtained from each patient included in the study. The study protocol conforms to the ethical guidelines of the 1975 Declaration of Helsinki as reflected in a priori approval by the Ethics Committee of HUCFF-UFRJ (APPROVAL NUMBER/ID: 3.080.459).

### 2.2. Transient Elastography (TE)

Transient elastography of the liver was performed with FibroScan^®^ Touch 502 equipment (Echosens, Paris, France), first using probe M. The XL probe was used sequentially in case of failure or unreliable results. Measures were acquired with the patient lying supine, with the right arm elevated to facilitate access to the right liver lobe. The probe tip contacted the intercostal skin with coupling gel. The entire procedure was considered to have failed when no value was obtained after ten shots. The final result of the TE measures was considered valid if the following criteria were fulfilled: ten valid measurements with a success rate (the ratio of accurate shots to the total number of shots) of at least 60% and an interquartile range (IQR, reflecting the variability of measurements) of 30% or lower. The median value was calculated and expressed in kilopascals (kPa). In order to discriminate between the stages of liver fibrosis, the following cutoffs were adopted according to Li et al.’s meta-analysis [18]: F0/1 ≤ 7.1 kPa; F2 ≥ 7.2 kPa and ≤ 9.3 kPa; F3 ≥ 9.4 kPa and ≤ 12.1 kPa; F4 ≥12.2 kPa. Steatosis was evaluated by the controlled attenuation parameter (CAP), and the results were expressed in decibels per meter (dB/m). The presence of steatosis was considered positive if CAP ≥ 275 dB/m [9].

### 2.3. Two-Dimensional Shear Wave Elastography (2D-SWE)

Two-dimensional shear wave elastography was performed using Aixplorer™ equipment (Supersonic Imagine, Aix-en-Provence, France). The convex probe (SC6-1) was used for all measurement acquisition. The 2D-SWE exam was performed in the supine position, with maximum abduction of the right arm, with the probe placed in the right lobe of the liver. Patients were instructed to keep their apnea for five seconds while each measure was taken. The operator chose the target area of the liver guided by a real-time B-mode image, placing the sample box approximately 3.5 cm × 2.5 cm in size 2–3 cm under the liver capsule, in an area of liver parenchyma free of large vessels. A circular region of interest (ROI) of 15 mm was placed within the center of the rectangular sample box, and the mean value of liver stiffness and IQR was displayed. The ROI could be adjusted to avoid liver parenchyma structures, obtain the most homogenous area and adapt to narrow intercostal spaces. Three 2D SWE images were obtained for each patient, and the median value of these measures was considered the final stiffness. The cutoff points suggested by Hermann’s meta-analysis [19] were adopted as follows: F0/F1 ≤ 7.0 kPa; F2 ≥ 7.1 kPa and ≤ 8.0 kPa; F3 ≥ 8.1 kPa and ≤ 11.4 kPa; and F4 ≥ 11.5 kPa.

### 2.4. Statistical Analysis

The statistical analysis was performed using SPSS, version 21 (IBM SPSS Statistics, V.24.0. Armonk, NY, USA). Numerical variables with normal distribution are expressed as mean values and standard deviations, while for nonparametric variables median values and interquartile ranges were used. The categorical variables are reported as absolute (*n*) and relative (%) frequencies. As necessary, the Chi-square test or Fisher's exact test was used in the comparative analysis among proportions. The Student's *t*-test and the Mann–Whitney U test were applied to compare numerical variables. Spearman’s test was used for correlation analysis of TE and 2D-SWE results. Kappa indexes were used to measure agreement/concordance between the estimates of liver fibrosis obtained via the two-elastography methods. Logistic regression analysis was used to identify the variables independently associated with the disagreement between methods (outcome variables). For the concordance analysis, we categorized the stages of liver fibrosis as F0/1 vs. F ≥ 2, F0/1-F2 vs. F ≥ 3 and F0/1-F2-F3 vs. F4. 

The co-variables included in the binary logistic regression analysis were those with clinical relevance, such as age, gender, BMI, systemic arterial hypertension, diabetes mellitus and antiviral treatment. The outcome was the discordance between both methods defined by any discordance observed among the three categories (F0/1 vs. F ≥ 2, F0/1-F2 vs. F ≥ 3 and F0/1-F2-F3 vs. F4). A *p*-value < 0.05 was considered significant. 

## 3. Results 

One hundred and fifty patients were enrolled in this study. Five patients were excluded from the study, four due to failure to obtain elastography measures with the 2D-SWE (all with BMIs > 30 kg/m^2^), and one patient was excluded after the diagnosis of alfa1-antitrypsin deficiency. All elastography measurements performed with TE were valid. Therefore, one hundred and forty-five patients with valid transient elastography and real-time shear wave elastography measurements were included in the final analysis. The main characteristics of the subjects included in this study are presented in Table 1. 

The success rate of 2D-SWE was 97.3% compared with the 100% success rate of TE. Liver stiffness measures ranged from 2.2 to 29.9 kPa for TE and 3.3 to 22.1 kPa for 2D-SWE. The mean liver stiffness value for TE was 5.5 ± 4.3 kPa and that for 2D-SWE was 5.4 ± 3.0 kPA. Regarding TE, the M probe was used to perform 107 (73.8%) exams and the XL probe was used to perform 38 (26.2%). The mean BMI of patients for TE performed with the M probe was 24.9 ± 3.3 kg/m^2^ compared to 32 ± 4.2 kg/m^2^ for those for whom the XL probe was used (*p*< 0.001). The distribution of liver fibrosis stages according to TE and 2D-SWE is presented in Table 1. Considering CAP ≥ 275 dB/m for the diagnosis of steatosis, 20.0% of patients had steatosis. 

One hundred and fifteen exams (79.3%) estimated the same fibrosis stage according to concordance with the established categorization. When categorizing fibrosis stages in significant fibrosis (F0/1 vs. F2-3-4), advanced fibrosis (F0-1-2 vs. F3-4) and cirrhosis (F0-1-2-3 vs. F4), there were 30 misclassifications between TE and 2D-SWE (Table 2). The correlation analysis is shown in Figure 1. There was no difference between M and XL probes (*p* = 0.527) or in the agreement between methods.

The Spearman’s correlation coefficient between both elastography methods was 0.71 (*p* < 0.01), indicating a strong correlation between liver stiffness measures obtained with TE and 2D-SWE. Kappa coefficient index of agreement according to the different categories of liver fibrosis by both TE and SWE is presented in Table 3.

Table 4 shows the agreement between TE and 2D-SWE in untreated patients, antiviral therapy patients, and patients with and without steatosis. In untreated patients, the agreement between the two methods was higher than in patients using antiviral therapy (0.76 vs. 0.47) and higher in patients without steatosis than those with steatosis (0.63 vs. 0.12). The Kappa coefficient was considered strong in untreated patients and those without steatosis, moderate in treated patients and weak in patients with steatosis. 

The main characteristics of the concordant and discordant exams are expressed in Table 5. The analysis of logistic regression models revealed that the variables independently associated with discordance between both methods were diabetes mellitus (OR 5.04; 95%CI: 1.89-13.3; *p* < 0.001) and antiviral treatment (OR 6.79; 95%CI: 2.33–19.83; *p* < 0.001).

## 4. Discussion

This study compared the agreement and correlation between TE and 2D-SWE in assessing liver fibrosis in patients with chronic hepatitis B and showed a strong correlation. The correlation between liver stiffness measures obtained with TE and 2D-SWE was similar to that observed by Zeng et al. [16], even though our study was characterized by a higher prevalence of obesity. In addition, it has also been shown for the first time that diabetes mellitus and the use of antiviral treatment were independently associated with disagreement in staging liver fibrosis with these two elastography methods. The overall coefficient of agreement of the exams was moderate. However, when the fibrosis stages were categorized, the agreement was substantial, especially in the F0/1 category, in which both methods were more accurate [9,19,20].

Few studies have compared the two methods in patients with chronic hepatitis B [9,10,11]. All previous studies comparing both methods were performed on an Asian HBV- naive population with a lower BMI than ours—57% of our population presented with overweight and obesity. Factors associated with disagreement between the methods were not evaluated, and other data related to metabolic syndrome were unavailable. In our study, BMI was associated with an inability to obtain reliable LSMs for 2D-SWE but not for TE. Our TE measures were valid and different from previous studies [14,15,16,21]. This could be attributed to our using the XL probe in obese patients, reducing the odds of invalid measures. Indeed, there was no difference observed in using M or XL probes when comparing the agreement and disagreement groups (*p* = 0.528). Using the XL probe increases the viability of elastography, especially in obese patients. The performance of the XL probe in differentiating between liver fibrosis stages is similar to that of the M probe [22,23]. There are no recommendations for using different cutoffs for M and XL probes. Our study used the XL probe to eliminate failure in obtaining liver stiffness measures with TE.

In our study, 2D-SWE failed to obtain elastography measures in four patients (2.7%), all of whom had BMIs > 30 kg/m^2^. This finding was similar to the results obtained by Jamialahmadi et al. [24]. The ideal distance from the ultrasound transducer to generate liver stiffness measures is 4.0–4.5 cm [25], which could explain unreliable results in obese patients. 

Other peculiarities of our population were the high prevalences of diabetes (21.4%) and metabolic syndrome (28.7%). Metabolic profile analysis is relevant to evaluating whether other factors could influence liver stiffness measures by elastography. Diabetes typically correlates with steatosis, high BMI and liver inflammation from NASH in some patients. As liver biopsies were not performed, we cannot attribute this result to inflammation caused by nonalcoholic steatohepatitis or improvement with antiviral treatment. However, these might be possible explanations for this finding. Additionally, we might hypothesize that patients with fatty liver did better with TE because the XL probe is unavailable for 2D-SWE. Interestingly, although diabetes was a source of disagreement between the two elastography methods, BMI was only associated with failure to obtain reliable measures with 2D-SWE, reinforcing the theory that using the XL probe might have had some impact on our results. 

These data were not evaluated in previous studies in which both methods were compared in patients with HCB. To the best of our knowledge, no other study has analyzed the factors associated with disagreement in staging liver fibrosis with TE and 2D-SWE in patients with chronic hepatitis B. 

Only three studies have compared both methods in patients with chronic hepatitis B [14,15,26]. In Leung et al.’s study, the accuracy of both methods in the staging of liver fibrosis was compared, and 2D-SWE had a stronger correlation and higher accuracy. The number of patients for whom real-time elastography over- or underestimated fibrosis stage was similar, with no differences in demographic or serologic profiles [14]. In the research conducted by Zeng et al., the methods presented a correlation similar to ours (*r* = 0.835 vs. *r* = 0.71), even though our study had higher proportions of overweight or obese patients, steatosis and use of antiviral therapy. In this study, the accuracy of the elastographic methods was similar [15]. The same finding was confirmed by Yao et al. [26].

The agreement between elastography methods for untreated patients was remarkably higher than for those receiving antiviral therapy. This could be attributed to the fact that most patients in the first group were in the chronic infection stage and therefore had no clinically relevant fibrosis or inflammation to interfere with liver stiffness measurements. Many studies have shown that liver stiffness declines after antiviral therapy in patients with CHB. However, this may not reflect fibrosis regression but the reduction in inflammation and ALT [27,28]. A recent systematic review demonstrated that the magnitude of the decline is incremental over time since the start of therapy and is higher in patients with high baseline stiffness and viral replication activity [29]. Studies on the change in liver stiffness measures with 2D-SWE in patients with CHB receiving antiviral therapy still need to be completed. 

Elevated ALT levels might affect the predictive accuracy of TE [30,31]; however, Cardoso et al. reported that TE cutoff values adjusted to ALT levels did not improve the performance of liver stiffness in CHB patients. Nonetheless, TE value increases with necroinflammatory activity and more significant hepatic steatosis [10]. A recent meta-analysis of TE is of great value for the detection of CHB-related cirrhosis; however, it has a suboptimal performance in detecting significant fibrosis. Further studies should focus on the TE cutoff value and the effect of ALT elevation in patients with CHB [32]. A study by Zhuang et al. showed that inflammation activity, ALT, AST and GGT can influence the acquisition of elastography measures with 2D-SWE [17].

The agreement between the two methods in patients without steatosis was remarkably more robust than in patients with steatosis. Only a few studies have explored the impact of steatosis and nonalcoholic fatty liver disease (NAFLD) in patients with chronic hepatitis B. In Cai et al.’s [33] study, severe steatosis by histology or CAP ≥287 dB/m by TE was independently linked to increased LSM values and false-positive rates in patients with low-stage fibrosis. This study concluded that steatosis might lead to the overestimation of fibrosis assessed by liver stiffness measurements in patients with chronic hepatitis B.

However, Zhang et al. [34] showed that the degree of hepatic steatosis might not influence the performance of TE in assessing liver fibrosis in patients with chronic hepatitis B. There was no significant difference in the correlation coefficient of TE with fibrosis stage among the different degrees of hepatic steatosis. These findings suggest that TE could be reliable even in concurrent chronic hepatitis B and NAFLD patients. The role of hepatic steatosis in chronic hepatitis B remains controversial, but evidence supports that metabolic syndrome is an independent factor of liver cirrhosis in CHB [35,36]. A study by Wong et al. demonstrated that, in patients with CHB, the metabolic syndrome components were more prevalent in patients with liver stiffness measures ≥ 13.4 kPa. This factor was independently associated with cirrhosis, not steatosis [35].

To date, no studies have evaluated the influence of steatosis and NAFLD in hepatic fibrosis staging in chronic hepatitis B using 2D-SWE. In our study, CAP did not impact the discordance between measures regarding liver fibrosis. 

The noteworthy finding of this study was that TE and 2D-SWE had good agreement and correlation in staging liver fibrosis in CHB patients. Diabetes mellitus and antiviral treatment were independently associated with disagreement in staging liver fibrosis with TE and 2D-SWE. 

This finding could be relevant for patients with CHB and diabetes in clinical practice. As elastography methods have poorer agreement in this group of patients, the same method should be used in follow-ups for liver stiffness measurements and fibrosis staging. Further studies must explain how antiviral therapy and diabetes influence elastography methods and elucidate the confounding factors in noninvasively staging liver fibrosis.

## 5. Conclusions

In conclusion, in patients with chronic hepatitis B, TE and 2D-SWE showed strong correlation and good agreement in identifying mild versus advanced fibrosis stages. Diabetes mellitus and antiviral therapy use might impact the agreement between liver stiffness measures obtained with these two different elastographic methods. Further studies are necessary to elucidate the mechanisms involved in the influence of diabetes and antiviral therapy in noninvasively staging liver fibrosis.

## Figures and Tables

**Figure 1 viruses-15-00846-f001:**
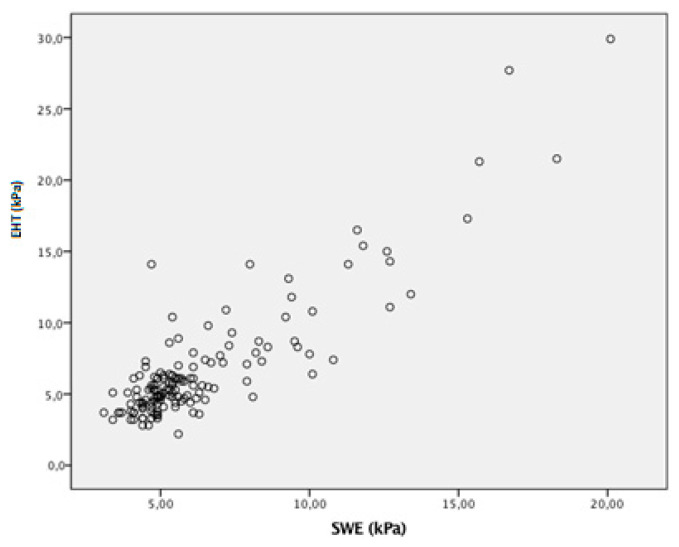
Scatter plot of distribution of transient elastography (TE) vs. 2D-shear wave elastography (2D-SWE).

**Table 1 viruses-15-00846-t001:** Demographic, serological and laboratorial data for patients with chronic hepatitis B (*n =* 145).

Parameter	
Male, *n* (%)	85 (58.6%)
Age (years) ^Ɨ^	48 ± 14
BMI (kg/m^2^) ^Ɨ^	26.71 ± 4.61
Overweight, *n* (%)	53 (35.6%)
Obese, *n* (%)	31 (21.4%)
Waist circumference (cm) ^Ɨ^	93.6 ± 11.2
AST (U/L) ^ǂ^	24 (20–30)
ALT (U/L) ^ǂ^	27 (19–39)
γGT (U/L) ^ǂ^	28 (18–41)
Total bilirubin (mg/dL) ^ǂ^	0.60 (0.49–0.97)
Serum albumin (g/L) ^Ɨ^	4.34 ± 0.49
Platelet count (10^9^/L) ^Ɨ^	200.4 ± 59.33
Hba1c (%) ^ǂ^	5.6 (5.2–5.9)
Total cholesterol (mg/dL) ^ǂ^	174 (151–205)
Triglycerides (mg/dL) ^ǂ^	89 (63–123)
LDL (mg/dL) ^ǂ^	101 (87–134)
Diabetes, *n* (%)	31 (21.4%)
Metabolic syndrome	39/136 (28.7%)
Chronic infection, *n* (%)	63 (43.5%)—all HBeAg (−)
Chronic hepatitis, *n* (%)	82 (56.6%)
	HBeAg (+): 5 (3.4%)
	HBeAg (−): 77 (53.1%)
Use of antiviral, *n* (%)	68 (46.9%)
	Tenofovir: 39 (26.9%)
	Entecavir: 26 (17.9%)
	Lamivudine: 3 (2.1%)
Steatosis evaluation	
CAP (dB/m) ^ǂ^	234 (200–265) [100–396]
CAP ≥ 275 dB/m	29 (20.0%)
TE fibrosis stage categorization, *n* (%)	F0/ F1: 73 (50.4%)
	F ≥ 2 = 40 (27.6%);
	F ≥ 3 = 21 (14.5%)
	F4 = 11 (7.6%).
2D-SWE fibrosis stages distribution, *n* (%)	F0/F1: 113 (77.9%)
	F2: 7 (4.8%)
	F3: 14 (9.7%)
	F4: 11 (7.6%)

^Ɨ^ Mean ± standard deviation. ^ǂ^ Median (interquartile range).

**Table 2 viruses-15-00846-t002:** Distribution of fibrosis stages in significant fibrosis (F0/1 vs. F2-3-4), advanced fibrosis (F0/1-2 vs. F3-4) and cirrhosis (Fo-1-2-3 vs. F4) according to TE and 2D-SWE.

			2D SWE		Total
			**F0/F1**	**F2-3-4**	
**F0/1 vs. F2-3-4**	**TE**	**F0/F1**	103	2	105
		**F2-3-4**	10	30	40
		**Total**	113	32	145
**F0-1-2 vs. F3-4**			**F0-1-2**	**F3-4**	
	**TE**	**F0-1-2**	115	9	124
		**F3-4**	5	16	21
		**Total**	120	25	145
**F0-1-2-3 vs. F4**			**F0-1-2-3**	**F4**	
	**TE**	**F0-1-2-3**	130	2	132
		**F4**	4	9	13
		**Total**	134	11	145

**Table 3 viruses-15-00846-t003:** Kappa coefficient index of agreement according to the different categories of liver fibrosis by both TE and SWE categorization.

Category	Kappa	*p*-Value
**Overall**	0.58	<0.01
**F0/1 vs. F2-3-4**	0.78	<0.01
**F0-1-2 vs. F3-4**	0.64	<0.01
**F0/1-2-3 vs. F4**	0.73	<0.01

**Table 4 viruses-15-00846-t004:** Kappa coefficient index of agreement between TE and 2D-SWE in patients not in treatment and patients receiving antiviral therapy, and in patients without and with steatosis.

Category	Kappa	*p*-Value
General (*n =* 145)	0.58	<0.01
Not in treatment (*n =* 77)	0.76	<0.01
Patients receiving antiviral therapy (*n =* 68)	0.47	<0.01
Without steatosis (*n =* 116)	0.63	<0.01
With steatosis (*n =* 29)	0.12	<0.01

**Table 5 viruses-15-00846-t005:** Summary of demographic data and blood test and elastography results for patients with concordant and discordant exams between TE and 2D-SWE.

Parameter	Concordant (*n* = 115)	Discordant (*n* = 30)	*p*-Value
No. of men (%)	64 (55.7)	21 (70.0)	0.155
No. of diabetic patients (%)	19 (17.1)	14 (48.3)	<0.001 *
No. of dyslipidemic patients (%)	30 (27.5)	13 (44.8)	0.074
No. of patients with hypertension (%)	19 (21.1)	10 (50.0)	0.008 *
No. of patients receiving antivirals (%)	44 (38.3%)	24 (80.0%)	<0.001 *
Chronic infection	57 (49.6%)	5 (16.7%)	0.010 *
Age (years) ^Ɨ^	46.8 ± 14.4	52.1 ± 13.7	0.193
Weight (kg) ^Ɨ^	75.2 ± 14.0	77.8 ± 16.3	0.460
BMI (kg/m^2^) ^Ɨ^	26.6 ± 4.5	27.2 ± 5.2	0.241
Waist circumference (cm) ^Ɨ^	92.9 ± 11.1	98.2 ± 9.8	0.031 *
Hba1c (%) ^Ɨ^	5.6 ± 1.0	6.2 ± 1.3	0.032 *
Albumin (g/dL) ^Ɨ^	4.3 ± 0.5	4.3 ± 0.4	0.431
Platelets ^Ɨ^	202.294 ± 58.133	190.290 ± 66.547	0.338
AST (IU/l) ^ǂ^	25 (20–30)	23 (21–32)	0.450
ALT (IU/l) ^ǂ^	28 (20–40)	27 (17–42)	0.259
γGT (IU/l) ^ǂ^	25 (17–39)	41 (25–59)	<0.001 *
Alkaline phosphatase (IU/l) ^ǂ^	79 (64–124)	110 (66–172)	0.095
Total cholesterol (mg/dL) ^ǂ^	172 (151–201)	195 (148–212)	0.906
HDL-cholesterol (mg/dL) ^ǂ^	50 (42–59)	45 (36–49)	0.129
LDL-cholesterol (mg/dL) ^ǂ^	100 (87–130)	95 (85–145)	0.654
Triglycerides (mg/dL) ^ǂ^	82 (60–111)	124 (100–170)	<0.001 *
TE (kPa) ^ǂ^	5.1 (4.3–6.1)	8.3 (6.4–10.4)	<0.001 *
CAP (dB/m) ^ǂ^	234 (206– 256)	256 (213–313)	0.035 *
M probe, *n* (%)	85 (73.9%)	21 (70.0%)	0.661
2D-SWE (kPa) ^ǂ^	5.3 (4.8–6.2)	6.6 (5.1–9.6)	<0.001 *

^Ɨ^ Mean ± standard deviation. ^ǂ^ Median (interquartile range). * *p* < 0.05.

## Data Availability

The datasets used and/or analyzed during the current study are available from the corresponding author on reasonable request.

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
