# Peer review of "Factors Associated with Disagreement of Fibrosis Stages between 2D-Shear Wave Elastography and Transient Elastography in Chronic Hepatitis B"

_viruses, 2023, doi:10.3390/v15040846_

Round 1
Reviewer 1 Report
The study by Gdalevici et al is a cross-sectional study between 2016 and 2019 evaluating the agreement between transient elastography (TE) and two-dimensional shear wave elastography (2D-SWE) in patients with chronic hepatitis B (CHB)(n=150). The authors found a strong correlation (Spearman =0.71) and good agreement (Kappa= 0.58) according to the previously published cutoffs that categorized the liver fibrosis stage.
However, the authors should make major changes to improve the quality of their results and to be accepted in viruses
Major comments:
1. Abstract. Please, include background.
2. Material and Methods.
a. None of the non-invasive techniques is capable of identifying contiguous stages of liver fibrosis, nor is LSM. Therefore, authors should recategorize their cutoffs to identify significant fibrosis (F2-4), advanced fibrosis (F2-3), and cirrhosis (F4) for TE (doi:10.1111/apt.13488) and 2D-SWE (doi: 10.1002/hep.29179)
b. Please, include references regarding the LSM with TE and with 2D-SWE.
c. Please, include references about the quality criteria of LSM with TE and with 2D-SWE
3. Statistical analysis.
a. Please, include antiviral treatment and the time of antiviral treatment as co-variables in the logistic regression analysis
b. The low number of discordant exams (n=25) limits the number of included variables in the multivariate analysis. Please, reconsider the models to avoid inaccurate or unreliable results.
4. Results.
a. Authors should include more information about the applicability of each technique. Please include the number of patients for each exclusion criterion in TE and 2D-SWE
b. Please, include the interpretation of results in the Discussion section but not in the results (ie lines 203-204)
5. Table 1. Please, include the treatment time for each antiviral drug and recategorize liver fibrosis stages.
6. Table 2 and table 3. Please, recategorize liver fibrosis stages as significant fibrosis (F2-4), advanced fibrosis (F3-4), and cirrhosis (F4)
7. Table 5. The low number of discordant exams (n=25) limits the number of included variables in the multivariate analysis.
8. Discussion.
a. Update References. Please, include and discuss important references as doi.org/10.1155/2018/3406789
doi.org/10.1016/j.dld.2018.05.005
doi.org/10.1155/2018/3406789
b. Please include the limitations of the study
Minor comments
· Line 212 “concordant and discordant”
· Abstract. Please, include the new fibrosis stage categorization
· Please, use two-dimensional shear wave elastography (2D-SWE) better than Real-time SWE
Reviewer 2 Report
The authors aimed to evaluate the non-invasive methods for liver fibrosis agreement between transient elastography (TE) and real-time 2D-shear wave elastography (2D-SWE) in patients with chronic hepatitis B. They analyzed the factors related to the disagreement of measures between these two elastography methods.
The study used large volume of CHB patients. However, there are some flaws in designing the experiments and did not show any significant differences between those methods, which makes no novelty in the study. In addition, research study could have been improved to delineate the mechanism of how the 2D-SWE is better than TE and there is no solid data showing that the disagreement factor was diabetes mellitus (DM). What is the role DM playing in liver fibrosis and are there any correlation between them in CHB patients. Introduction can be improved with updated literatures and relevant articles. English corrections are needed. Several reports were already published about non-invasive methods for liver fibrosis and this study doesn't correlate with the results derived and conclusion.
Author Response
Wynne Wang and Doina Virlan
Editors in Chief
Viruses
February 24th, 2023
To the review Board of Viruses
Dear Dr. Wynne Wang and Dr. Doina Virlan
We thank you for the opportunity to have our manuscript DIABETES REDUCES AGREEMENT BETWEEN 2D-SHEAR WAVE ELASTOGRAPHY AND TRANSIENT ELASTOGRAPHY IN CHRONIC HEPATITIS B considered for publication in Viruses. We want to thank the Editors and Reviewers of Viruses for their kind and valuable comments and suggestions that helped improve the quality of our manuscript.
We read the comments carefully and answered all with the respective modifications made in the body of the text. All changes are highlighted in the manuscript as demanded. We look forward to our revised manuscript being accepted for publication in Viruses.
Best regards,
Cristiane Villela-Nogueira, on behalf of the authors
# Reviewer 2
First, we thank the reviewer for the essential comments and questions. We have updated our references, and the new cutoffs did not alter the final result.
This study evaluated factors associated with the disagreement between two different elastography methods. Also, a liver biopsy was not performed to help elucidate the mechanisms involved in the discordance of exams. We included the lack of liver biopsy as one of the study's limitations. However, as liver biopsy is an invasive test and since it would not be indicated in our patients for management decisions, it would not be adequate to perform a liver biopsy to investigate additional mechanisms to explain the discordance of the elastographic techniques.
Please see the attachment

Round 2
Reviewer 1 Report
viruses-2147635
Diabetes reduces agreement between 2D-Shear Wave Elastography and Transient Elastography in chronic hepatitis B
Comments to Authors:
The study by Gdalevici et al is a cross-sectional study between 2016 and 2019 evaluating the agreement between transient elastography (TE) and two-dimensional shear wave elastography (2D-SWE) in patients with chronic hepatitis B (CHB)(n=150). The authors found a strong correlation (Spearman =0.71) and good agreement (Kappa= 0.58) according to the previously published cutoffs that categorized the liver fibrosis stage.
Authors should make major changes to improve the quality of their results and to be accepted in viruses
Major comments:
It is very important that authors do not use the non-invasive techniques as a liver biopsy, because none of the non-invasive techniques can identify contiguous stages as individual stages. Authors should use TE and 2D-SWE cut-offs to identify F2-3-4 (significant fibrosis) and F3-4 (advanced fibrosis) but not F2 or F3.
Remember that F2 (portal fibrosis) is not the same as F2-3-4 (significant fibrosis) and F3 (periportal fibrosis) is not the same as F3-4 (advanced fibrosis).
Therefore, the Authors should use only the accepted categorized fibrosis stages: significant fibrosis (F2-4), advanced fibrosis (F2-3), and cirrhosis (F4) for TE (doi:10.1111/apt.13488) and 2D-SWE (doi: 10.1002/hep.29179) in all their study (abstract, methods, results, tables and discussion section). Despite the recategorization did not alter the final result this is the most accurate form to show the results.
Reviewer 2 Report
The author can include some of recent literatures and minor text corrections.
Author Response
Thank you for the opportunity to consider our manuscript DIABETES REDUCES AGREEMENT BETWEEN 2D-SHEAR WAVE ELASTOGRAPHY AND TRANSIENT ELASTOGRAPHY IN CHRONIC HEPATITIS B. We would like thank very much the Editor and Reviewers of Viruses for their kind and useful comments and suggestions that surely helped to improve the quality of our manuscript.
We read the comments carefully and answered all with the respective modifications made in the body of the text. All changes are highlighted in the manuscript as demanded. We are looking forward to having our revised manuscript accepted for publication in Viruses.
Best regards,
Cristiane Villela-Nogueira, on behalf of the authors
Reviewer comments
First, we would like to thank the reviewer for the important comments and questions. We have updated our references. All changes are highlighted in the manuscript as demanded.